# Distinct roles of striatal direct and indirect pathways in value-based decision making

Shinae Kwak[1,2], Min Whan Jung[1,2]*

[1]Center for Synaptic Brain Dysfunctions, Institute for Basic Science, Daejeon, Republic of Korea; [2]Department of Biological Sciences, Korea Advanced Institute of Science and Technology, Daejeon, Republic of Korea

**Abstract** The striatum is critically involved in value-based decision making. However, it is unclear how striatal direct and indirect pathways work together to make optimal choices in a dynamic and uncertain environment. Here, we examined the effects of selectively inactivating D1 receptor (D1R)- or D2 receptor (D2R)-expressing dorsal striatal neurons (corresponding to direct- and indirect-pathway neurons, respectively) on mouse choice behavior in a reversal task with progressively increasing reversal frequency and a dynamic two-armed bandit task. Inactivation of either D1R- or D2R-expressing striatal neurons impaired performance in both tasks, but the pattern of altered choice behavior differed between the two animal groups. A reinforcement learning model-based analysis indicated that inactivation of D1R- and D2R-expressing striatal neurons selectively impairs value-dependent action selection and value learning, respectively. Our results suggest differential contributions of striatal direct and indirect pathways to two distinct steps in value-based decision making.
DOI: https://doi.org/10.7554/eLife.46050.001

*For correspondence:
mwjung@kaist.ac.kr

Competing interests: The authors declare that no competing interests exist.

## Introduction

The striatum is critically involved in value-based decision making, which consists of two distinct steps: value-based action selection and value updating based on choice outcomes. A large body of evidence indicates the involvement of the striatum in both of these processes (*Balleine et al., 2007*; *Hikosaka et al., 2006*; *Ito and Doya, 2011*; *Lee et al., 2012*; *Macpherson et al., 2014*). Striatal spiny projection neurons (SPNs) are divided into two distinct groups according to their output projections. In rodents, direct-pathway SPNs project directly to the endopeduncular nucleus (EP; homologous to the globus pallidus interna in primates) and the substantia nigra pars reticulata (SNr), and indirect-pathway SPNs project indirectly to the EP/SNr via the globus pallidus (GP) and subthalamic nucleus (*Smith et al., 1998*). The two groups of striatal neurons also differ in their gene expression patterns. Direct-pathway striatal neurons selectively express D1 receptors (D1R), whereas indirect-pathway striatal neurons express D2 receptors (D2R) (*Gerfen et al., 1990*), although such a segregation is less strict in the ventral striatum (*Smith et al., 2013*). Selectively manipulating striatal D1R (or direct-pathway SPNs) versus D2R (or indirect-pathway SPNs) affects reward-based learning and goal-directed behavior differently (*Hikida et al., 2010*; *Kravitz et al., 2012*; *Lee et al., 2015*; *Nakamura and Hikosaka, 2006*; *Nonomura et al., 2018*; *Tai et al., 2012*; *Yawata et al., 2012*), suggesting distinct roles of direct and indirect pathways in value-based decision making. However, it is unclear how the direct and indirect pathways of the striatum work together to control value-based action selection and value updating. In the present study, to obtain insights on the roles of striatal direct and indirect pathway neurons in these processes, we selectively inactivated D1R- or D2R-expressing dorsal striatal neurons and examined subsequent effects on mouse choice behavior in reversal and dynamic foraging tasks. We found a double dissociation in the effects of inactivating D1R- versus D2R-expressing striatal neurons: D1R neuronal inactivation reduced value-dependent

action selection and D2R neuronal inactivation reduced value learning, with neither affecting the other process. Our results indicate that D1R- and D2R-expressing dorsal striatal neurons are indispensable for two different steps in value-based decision making.

## Results

### Reversal task

We used mice harboring a D1R-Cre or D2R-Cre construct to selectively inactivate D1R- or D2R-expressing striatal neurons, respectively. We bilaterally injected a double-floxed (DIO) Cre-dependent adeno-associated virus (AAV) vector carrying a modified form of the human M4 muscarinic receptor (DIO-hM4Di-mCherry) into the dorsal striatum of 31 D1R-Cre and 30 D2R-Cre mice. As controls, we bilaterally injected AAV virus carrying enhanced green fluorescent protein (DIO-eGFP) into the dorsal striatum of separate groups of D1R-Cre and D2R-Cre mice (n = 5 each). Histological examinations after completion of behavioral experiments revealed that mCherry and eGFP were expressed in the dorsal striatum and EP in D1R-Cre mice and in the dorsal striatum and GP in D2R-Cre mice, confirming their selective expression in direct- or indirect-pathway SPNs, respectively (*Figure 1*).

Twenty or 21 d after virus injection, D1R-Cre (n = 26) and D2R-Cre (n = 26) mice were trained in a reversal task in an operant chamber with progressively increasing reversal frequency. This was a self-paced instrumental learning task in which the animal initiates a trial by poking its nose into the central hole and then chooses freely either the left or right nose-poke hole to obtain a water reward (*Figure 2a*). D1R-Cre and D2R-Cre mice were each divided into three groups—CNO group, in which clozapine-N-oxide (CNO) was injected into hM4Di-expressing mice; DMSO group, in which dimethyl sulfoxide (DMSO, vehicle) was injected into hM4Di-expressing mice; and eGFP-CNO group, in which CNO was injected into eGFP-expressing mice (D1R-Cre mice, n = 11, 10 and 5 for CNO, DMSO and eGFP-CNO groups, respectively; D2R-Cre mice, n = 11, 10 and 5 for CNO, DMSO and eGFP-CNO

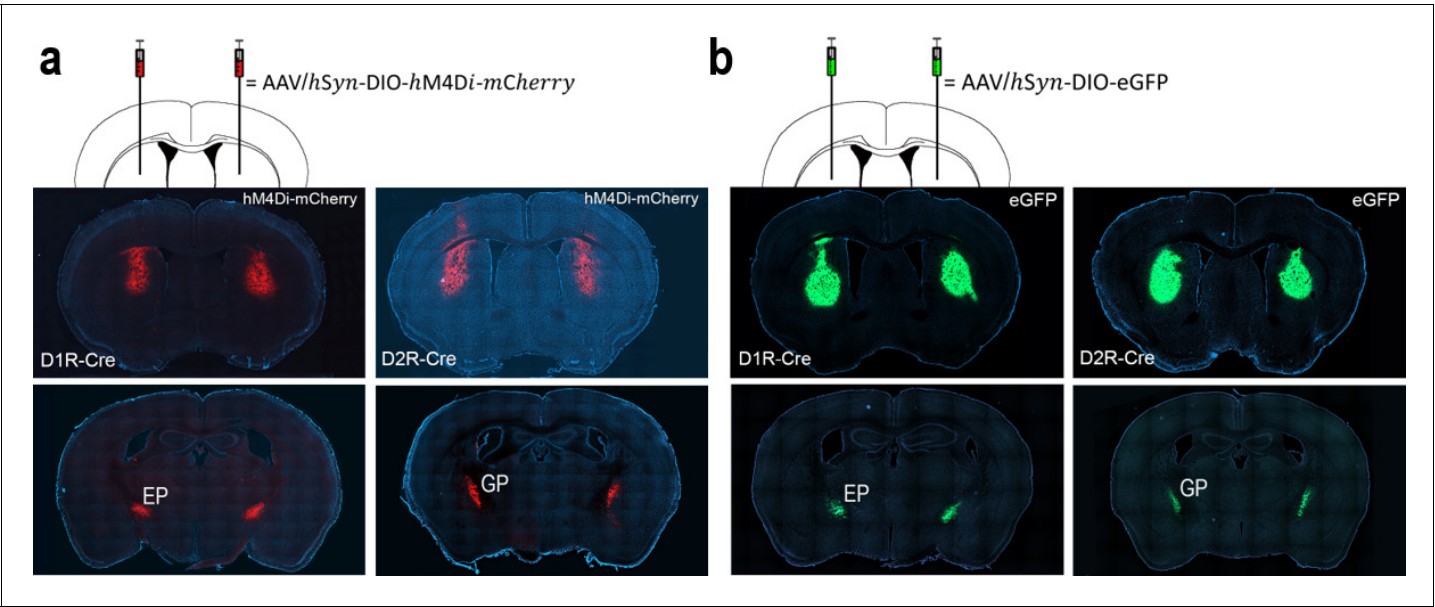

**Figure 1.** Selective expression of h4MDi-mCherry and eGFP in direct- or indirect-pathway striatal neurons. (**a**) Representative brain sections showing h4MDi-mCherry expression in the dorsal striatum and EP in D1R-Cre mice (left), and in the dorsal striatum and GP in D2R-Cre mice (right). (**b**) Representative brain sections showing eGFP expression in the dorsal striatum and EP in D1R-Cre mice (left), and in the dorsal striatum and GP in D2R-Cre mice (right).

DOI: https://doi.org/10.7554/eLife.46050.002

The following figure supplement is available for figure 1:

**Figure supplement 1.** Expression of D1 and D2 receptors in striatal interneurons.

DOI: https://doi.org/10.7554/eLife.46050.003

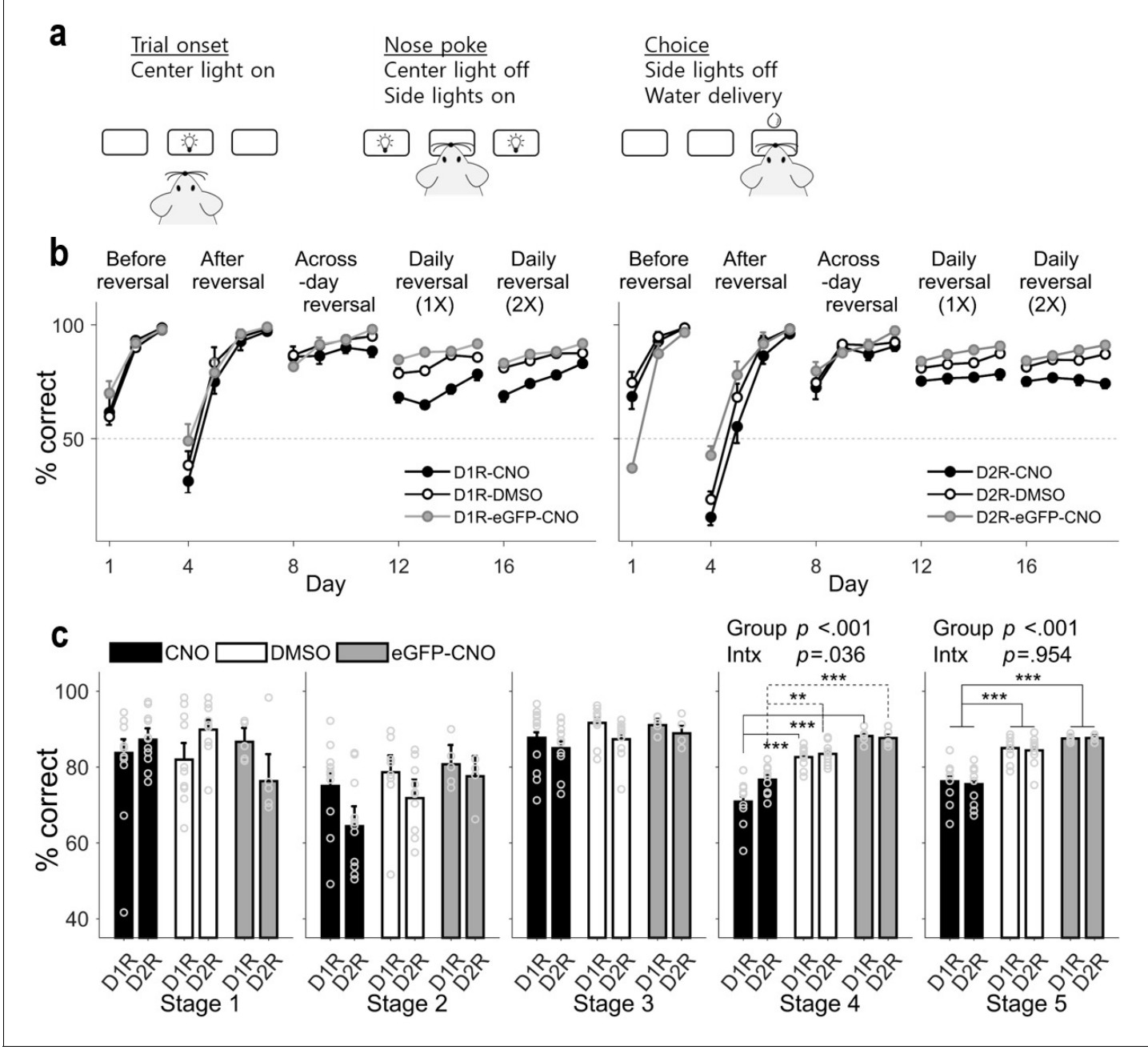

**Figure 2.** Behavioral performance in the reversal task. (**a**) Behavioral task. Following nose poke in the lit central hole, the animal was allowed to choose freely between two targets to obtain a water reward. (**b**) Daily performances (means ± SEM across animals) of the three animal groups (CNO, DMSO and eGFP-CNO) are shown separately for D1R-Cre and D2R-Cre mice. (**c**) Mean (± SEM across animals) performances of the three animal groups during each stage. Circles, data for individual animals. *P*-values are indicated for the main effect of animal group (Group) and the effect of mouse line ×animal group interaction (Intx) (two-way between-groups ANOVA). Asterisks indicate the results of Bonferroni post-hoc tests (**p<0.01; ***p<0.001).
DOI: https://doi.org/10.7554/eLife.46050.004

The following figure supplement is available for figure 2:

**Figure supplement 1.** CNO effects in the reversal task cannot be accounted for by differences in trial duration.
DOI: https://doi.org/10.7554/eLife.46050.005

groups, respectively). Mice were intraperitoneally injected daily with DMSO (2.5–3%, 0.5 ml/kg) or CNO (5 mg/kg) 40 min prior to behavioral testing. For each training stage and each animal group, sessions with mean trial durations greater than three standard deviations (SDs) from the mean of the corresponding population (i.e., trial-duration outliers) were excluded from the analysis (total deleted sessions: D1R-Cre mice, 4 of 190 DMSO sessions, 7 of 209 CNO sessions and 0 of 95 eGFP-CNO sessions; D2R-Cre mice: 3 of 190 DMSO sessions, 5 of 209 CNO sessions and 1 of 95 eGFP-CNO sessions).

All animal groups learned to choose the rewarding target (either left or right; counterbalanced across animals) during the initial 3 d of training (stage 1). After reversal of the rewarding target, all animal groups learned to choose the other rewarding target over 4 d of training (stage 2). Performances of the three animal groups were also similar during across-session reversal (reversal of the rewarding target at the beginning of each daily session; stage 3; two-way between-groups ANOVA, main effect of mouse line, $F_{(1,46)} = 2.8$, $p = 0.101$; main effect of animal group, $F_{(2,46)} = 2.04$, $p = 0.141$; mouse line × animal group interaction, $F_{(2,46)} = 0.15$, $p = 0.858$). However, performance of the CNO group was significantly lower compared with other animal groups in stage 4 (main effect of mouse line, $F_{(1,46)} = 3.64$, $p = 0.063$; main effect of animal group, $F_{(2,46)} = 58.54$, $p = 2.3 \times 10^{-13}$; mouse line × animal group interaction, $F_{(2,46)} = 3.58$, $p = 0.036$; post-hoc Bonferroni test, D1R-Cre mice, CNO vs. DMSO, $p = 9.9 \times 10^{-8}$, CNO vs. eGFP-CNO, $p = 2.1 \times 10^{-8}$, DMSO vs. eGFP-CNO, $p = 0.089$; D2R-Cre mice, CNO vs. DMSO, $p = 0.003$, CNO vs. eGFP-CNO, $p = 3.6 \times 10^{-5}$, DMSO vs. eGFP-CNO, $p = 0.326$) and stage 5 (main effect of mouse line, $F_{(1,46)} = 0.11$, $p = 0.747$; main effect of animal group, $F_{(2,46)} = 39.32$, $p = 1.1 \times 10^{-10}$; mouse line × animal group interaction, $F_{(2,46)} = 0.05$, $p = 0.954$; post-hoc Bonferroni test, CNO vs. DMSO, $p = 2.1 \times 10^{-8}$, CNO vs. eGFP-CNO, $p = 2.9 \times 10^{-9}$, DMSO vs. eGFP-CNO, $p = 0.195$; *Figure 2b, c*), in which the target location was changed one or two times during each daily session (within-session reversal). Mean trial durations varied substantially so that it was difficult to match them across the three animal groups. However, further analysis revealed that the performance deficits following D1R or D2R neuronal inactivation could not be attributed to differences in trial duration across animal groups (*Figure 2—figure supplement 1*). In sum, mice in all groups learned to choose the rewarding target as long as the location of the rewarding target did not change within a session. However, as the frequency of reversal increased such that the target location changed within a session, the performance of the CNO group became significantly impaired relative to that of the other groups.

## Dynamic two-armed bandit task

Having established that inactivation of D1R- or D2R-expressing striatal neurons impairs reversal learning and that this is not attributable to a nonspecific effect of CNO using separate groups of animals, we examined effects of inactivating D1R- or D2R-expressing striatal neurons in a dynamic two-armed bandit (TAB) task (*Figure 3a*) by injecting the same animal with DMSO and CNO on alternative days prior to daily sessions (10 sessions each). This allowed us to make within-subject comparisons between the effects of CNO and DMSO injection. We used some animals (five CNO and five DMSO D1R-Cre mice; five CNO and five DMSO D2R-Cre mice) that had been used in the reversal task (>7 d interval between the two tasks) as well as naive animals (10 D1R-Cre and nine D2R-Cre mice). For each treatment group, sessions with mean trial durations > 3 SDs from the mean of the corresponding treatment group were excluded from the analysis (deleted sessions: D1R-Cre mice, 3 of 200 DMSO sessions and 2 of 200 CNO sessions; D2R-Cre mice: 4 of 190 DMSO sessions and 6 of 190 CNO sessions). In addition, mean trial durations were matched between CNO and DMSO sessions by deleting long trial-duration CNO sessions (D1R-Cre mice, 24 of 198; D2R-Cre mice, 17 of 184) and short trial-duration DMSO sessions (D1R-Cre mice, 20 of 197; D2R-Cre mice, 4 of 186).

In the dynamic TAB task, each choice was associated with a different probability of reward that was kept constant within a block of trials, but changed across blocks without any sensory cues. Hence, the task required the animal to discover reward probabilities and the optimal choice based on the history of past choices and their outcomes. As shown previously in a similar TAB task (*Jeong et al., 2018*), the probability of choosing the lower-reward–probability target did not increase as a block transition approached, arguing against the possibility that animals were able to estimate the time of reversal (*Figure 3—figure supplement 1*). Consistent with this finding, animal choice behavior in this task was well captured by the Q-learning model, a simple reinforcement

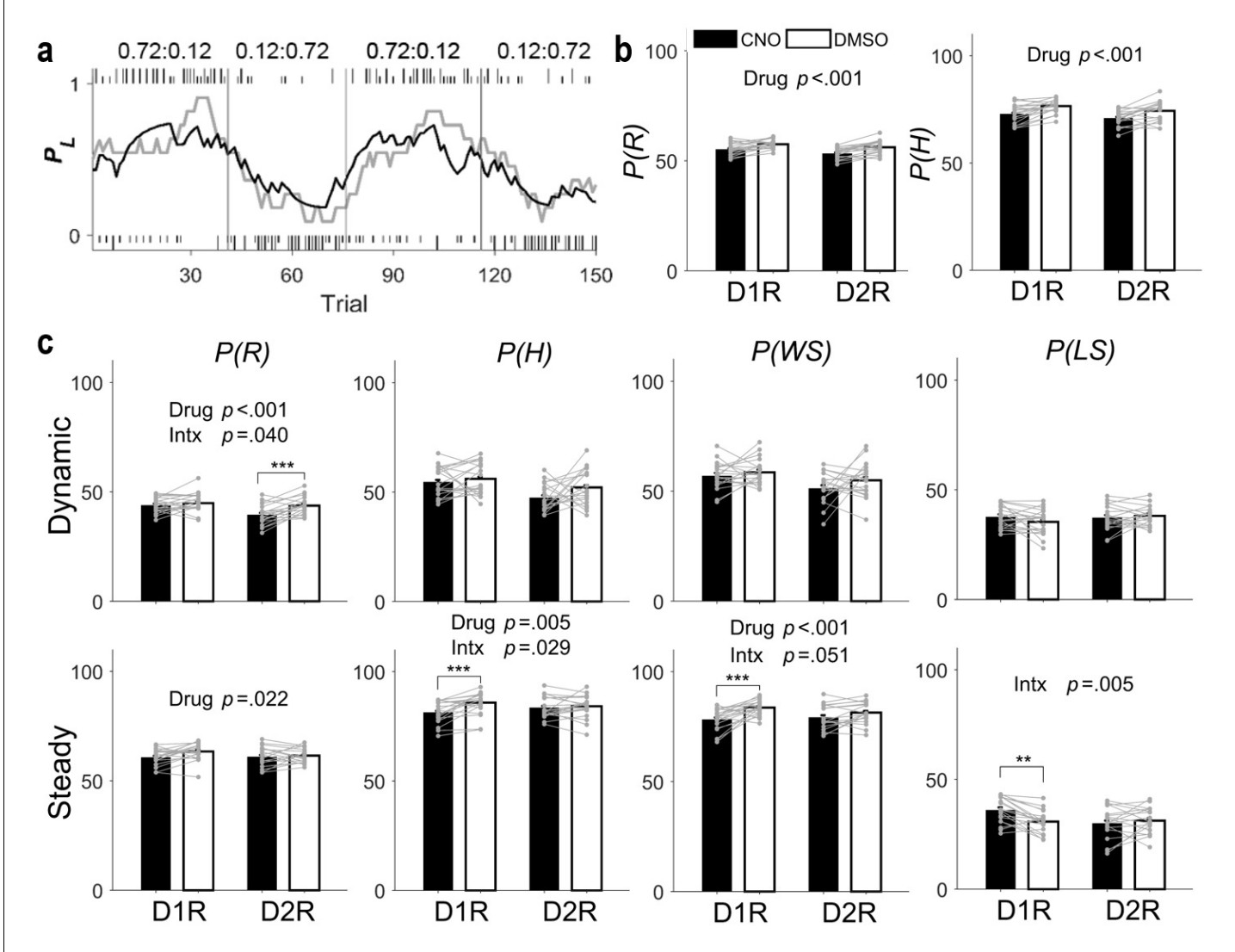

**Figure 3.** Behavioral performance in the dynamic TAB task. (a) Representative TAB-task session (D1R-Cre mouse with DMSO injection). Tick marks indicate trial-by-trial choices of the animal (top, left choice; bottom, right choice; long, rewarded; short, unrewarded). Gray vertical lines denote block transitions. Numbers indicate block reward probabilities of left and right targets. The gray line indicates actual choices of the animal, shown as the probability of choosing the left goal ($P_L$) in a moving average of 10 trials. The black line indicates $P_L$ predicted by the Q-learning model. (b–d) Proportions (%) of rewarded trials (P(R)), higher-reward–probability target choices (P(H)), win-stay (P(WS)), and lose-switch (P(LS)) were compared between DMSO and CNO sessions for all trials (b), dynamic-state trials (c), and steady-state trials (d) (means ± SEM across animals). Gray circles and connecting lines, individual animal data. *P*-values are indicated for those measures with significant main effects of drug and/or mouse line × drug interaction (Intx) effects (two-way mixed-design ANOVA). Asterisks indicate the results of Bonferroni post-hoc tests (**p<0.01; ***p<0.001).

DOI: https://doi.org/10.7554/eLife.46050.006

The following figure supplement is available for figure 3:

**Figure supplement 1.** Analysis results related to anticipating the time of block transition.

DOI: https://doi.org/10.7554/eLife.46050.007

learning model (*Sutton and Barto, 1998*) (*Figure 3a*). All animals were over-trained in the TAB task (~3 wk) before drug injection. CNO and DMSO were then injected on alternate days with either drug injected on the first day (counterbalanced across animals). We assessed behavioral performance by examining the proportions of rewarded trials and higher-reward–probability target choices (P(R) and P(H), respectively). We found a significant main effect of drug without a significant mouse line × drug interaction effect on P(R) (two-way mixed-design ANOVA, main effect of mouse line, $F_{(1,37)} = 6.214$, p = 0.017; main effect of drug, $F_{(1,37)} = 32.636$, p = $1.5 \times 10^{-6}$; mouse line × drug

interaction, $F_{(1,37)} = 0.530$, p = 0.471) as well as P(H) (main effect of mouse line, $F_{(1,37)} = 4.020$, p = 0.052; main effect of drug, $F_{(1,37)} = 31.591$, p = $2.1 \times 10^{-6}$; mouse line $\times$ drug interaction, $F_{(1,37)} = 0.0004$, p = 0.984; *Figure 3b*). These results indicate that inactivating either D1R- or D2R-expressing striatal neurons impairs performance in the TAB task.

To explore how D1R- and D2R-expressing neuronal inactivation impairs performance in the TAB task, we examined whether CNO effects differ between D1R-Cre and D2R-Cre mice. For this, we separately analyzed animal choice behavior in the dynamic and steady states (early and late trials after block transition, respectively; see Materials and methods) between which relative contributions of two major processes of value-based decision making, namely value-updating and value-dependent action-selection, to choice behavior are likely to vary. We examined whether choice-related measures, P(R) and P(H) along with the proportions of win-stay and lose-switch (P(WS) and P(LS), respectively), show significant mouse line $\times$ drug interaction effects in the dynamic or steady state. In the dynamic state, we found a significant mouse line $\times$ drug interaction effect on P(R) (two-way mixed-design ANOVA, main effect of mouse line, $F_{(1,37)} = 6.719$, p = 0.014; main effect of drug, $F_{(1,37)} = 14.76$, p = $4.6 \times 10^{-4}$; mouse line $\times$ drug interaction, $F_{(1,37)} = 4.5128$, p = 0.040), but not on the other measures (P(H), main effect of mouse line, $F_{(1,37)} = 10.469$, p = 0.003; main effect of drug, $F_{(1,37)} = 2.636$, p = 0.113; mouse line $\times$ drug interaction, $F_{(1,37)} = 1.0276$, p = 0.317; P(WS), main effect of mouse line, $F_{(1,37)} = 2.216$, p = 0.145; main effect of drug, $F_{(1,37)} = 3.4168$, p = 0.073; mouse line $\times$ drug interaction, $F_{(1,37)} = 0.19866$, p = 0.658; P(LS), main effect of mouse line, $F_{(1,37)} = 0.157$, p = 0.694; main effect of drug, $F_{(1,37)} = 0.6003$, p = 0.443; mouse line $\times$ drug interaction, $F_{(1,37)} = 2.5894$, p = 0.116). Post-hoc Bonferroni tests revealed a significant CNO effect on P(R) in D2R-Cre, but not D1R-Cre, mice (p = $1.8 \times 10^{-4}$ and 0.226, respectively; *Figure 3c*).

In the steady state, we found significant mouse line $\times$ drug interaction effects on P(H) (main effect of mouse line, $F_{(1,37)} = 0.009$, p = 0.926; main effect of drug, $F_{(1,37)} = 9.0145$, p = 0.005; mouse line $\times$ drug interaction, $F_{(1,37)} = 5.1513$, p = 0.029) and P(LS) (main effect of mouse line, $F_{(1,37)} = 0.853$, p = 0.362; main effect of drug, $F_{(1,37)} = 2.9786$, p = 0.093; mouse line $\times$ drug interaction, $F_{(1,37)} = 8.735$, p = 0.005), but not on P(R) (main effect of mouse line, $F_{(1,37)} = 0.921$, p = 0.344; main effect of drug, $F_{(1,37)} = 5.6916$, p = 0.022; mouse line $\times$ drug interaction, $F_{(1,37)} = 1.6644$, p = 0.205) or P(WS) (main effect of mouse line, $F_{(1,37)} = 0.094$, p = 0.761; main effect of drug, $F_{(1,37)} = 13.05$, p = $9.0 \times 10^{-4}$; mouse line $\times$ drug interaction, $F_{(1,37)} = 4.0786$, p = 0.051). Post-hoc Bonferroni tests revealed significant CNO effects on P(H) and P(LS) in D1R-Cre, but not D2R-Cre, mice (P(H), p = $5.6 \times 10^{-4}$ and 0.612 in D1R-Cre and D2R-Cre mice, respectively; P(LS), p = 0.002 and 0.396, respectively). Because the effect of mouse line $\times$ drug interaction on P(WS) was near the conventional criterion for significance (p = 0.051), we also performed post-hoc tests for this measure. CNO effect on P(WS) was significant in D1R-Cre, but not D2R-Cre, mice (p = $2.6 \times 10^{-4}$ and 0.273, respectively; *Figure 3c*). In sum, we found CNO effects that are selective between D1R-Cre and D2R-Cre mice for some behavioral measures. CNO significantly decreased P(R) in D2R-Cre, but not D1R-Cre, mice in the dynamic state, and significantly decreased P(H), P(WS) and P(LS) in D1R-Cre, but not D2R-Cre, mice in the steady state. To test the likelihood of finding three or more significant interaction effects by chance, we randomly assigned D1-Cre and D2R-Cre mice into two animal groups and repeated the same analysis (total eight ANOVAs; P(R), P(H), P(WS) and P(LS) in the dynamic and steady states). Out of 100 such permutations, we found no case in which significant animal group $\times$ drug interaction effect was found in three or more ANOVAs, indicating that our finding is unlikely to be obtained by chance.

## Model-based analysis

Differences in the pattern of CNO effects on animal choice behavior during dynamic and steady states between D1R-Cre and D2R-Cre mice raises the possibility that D1R- and D2R-expressing striatal neurons may contribute differently to the neural processes underlying value-based decision making. To further explore this possibility, we analyzed animal-choice data using the Q-learning model, a reinforcement learning model that has two free parameters: learning rate ($\alpha$) and randomness in action selection ($\beta$). The former determines the extent to which newly acquired information overrides old information, and the latter determines the degree of value-dependent action selection. We found that CNO significantly increased the randomness in action selection (or decreased value-dependent action selection) in D1R-Cre, but not D2R-Cre, mice (two-way mixed-design ANOVA,

main effect of mouse line, F(1,37) = 0.398, p = 0.532; main effect of drug, F(1,37) = 8.8886, p = 0.005; mouse line × drug interaction, F(1,37) = 7.2601, p = 0.011; post-hoc Bonferroni test, CNO vs. DMSO, D1R-Cre mice, p = 2.4 × 10$^{-4}$, D2R-Cre mice, p = 0.842). We also found that CNO significantly decreased learning rate in D2R-Cre, but not D1R-Cre, mice (main effect of mouse line, F (1,37) = 1.303, p = 0.261; main effect of drug, F(1,37) = 6.4289, p = 0.016; mouse line × drug interaction, F(1,37) = 5.9142, p = 0.020; post-hoc Bonferroni test, CNO vs. DMSO, D1R-Cre mice, p = 0.941, D2R-Cre mice, p = 0.001; *Figure 4a*). These results were consistent across several variants of the Q-learning model containing additional parameters (*Figure 4—figure supplement 1*; see *Supplementary file 1* for results of model comparisons). Collectively, these findings indicate that inactivation of D1R-expressing striatal neurons selectively impairs value-dependent action selection and inactivation of D2R-expressing striatal neurons selectively impairs value learning.

We also tested predictions of the above findings. In D2R-Cre mice, CNO is expected to slow the rate of action value change across trials after block transition because learning rate is reduced. However, during late trials after block transition (i.e., after sufficient learning), the magnitude of action values should be similar between CNO- and DMSO-injected sessions. In D1R-Cre mice, the effect of CNO on action value is expected to be weak because any effect of CNO on action value would be only indirect via its effect on action selection. To test these predictions, we compared action values for high- and low-probability reward targets ($Q_{high}$ and $Q_{low}$, respectively) between CNO and DMSO sessions for the initial 15 trials after block transition and the last 10 trials before block transition. We used blocks 2–4 for this analysis. $Q_{high}$ changed more slowly after block transition in CNO than DMSO sessions such that the mean $Q_{high}$ value in the dynamic state was significantly smaller in CNO compared with DMSO sessions in D2R-Cre, but not D1R-Cre, mice (two-way mixed-design ANOVA, main effect of mouse line, F(1,37) = 12.133, p = 0.001; main effect of drug, F(1,37) = 5.5095, p = 0.024; mouse line × drug interaction, F(1,37) = 14.147, p = 5.8 × 10$^{-4}$; post-hoc Bonferroni test, CNO vs. DMSO, D1R-Cre mice, p = 0.318, D2R-Cre mice, p = 1.3 × 10$^{-4}$; *Figure 4b*). In D1R-Cre mice, $Q_{high}$ was slightly higher during a few trials after block transition in CNO sessions compared with DMSO sessions (*Figure 4b*), but this can be explained by less value-dependent action selection in CNO sessions (i.e., greater chance of choosing the higher-reward–probability target immediately after block transition), which would increase the chance of updating the action value of the higher-reward–probability target. No significant effect of CNO was found on $Q_{low}$ after block transition (main effect of mouse line, F(1,37) = 0.299, p = 0.588; main effect of drug, F(1,37) = 0.2265, p = 0.637; mouse line × drug interaction, F(1,37) = 0.2891, p = 0.594; *Figure 4b*), suggesting preferential contributions of striatal D2R-expressing neurons to learning from positive outcomes (*Bayer and Glimcher, 2005*; *Fiorillo, 2013*) (see also *Figure 4—figure supplement 1*). As expected, $Q_{high}$ and $Q_{low}$ were similar between CNO and DMSO sessions in the steady state in D1R-Cre as well as D2R-Cre mice ($Q_{high}$, main effect of mouse line, F(1,37) = 4.881, p = 0.033; main effect of drug, F(1,37) = 0.54679, p = 0.464; mouse line × drug interaction, F(1,37) = 0.57919, p = 0.452; $Q_{low}$, main effect of mouse line, F(1,37) = 1.786, p = 0.190; main effect of drug, F(1,37) = 0.64447, p = 0.427; mouse line × drug interaction, F(1,37) = 0.41336, p = 0.524; *Figure 4c*). These results are consistent with the possibility that inactivation of D1R-expressing striatal neurons selectively impairs value-dependent action selection, whereas inactivation of D2R-expressing striatal neurons impairs value learning.

## Discussion

We found D1R neuronal inactivation decreases the degree of value-dependent action selection without affecting learning rate, whereas D2R neuronal inactivation decreases learning rate without affecting value-dependent action selection. These findings suggest that dorsal striatal direct and indirect pathways might play crucial roles in distinct stages of value-based decision making. Even though we did not test eGFP-CNO mice in the dynamic foraging task, selective effects of CNO on D1R-Cre versus D2R-Cre mice (as opposed to common CNO effects on both mouse lines) argue against non-specific effects of CNO. We also failed to find nonspecific effects of CNO on learning rate or randomness in action selection in our previous study (*Jeong et al., 2018*). There remains a possibility that inactivation of D1R- and/or D2R-expressing striatal interneurons (GABAergic fast-spiking interneurons and cholinergic tonically active neurons) might have contributed to the observed behavioral effects. However, this is unlikely because similar, small percentages of parvalbumin-positive and

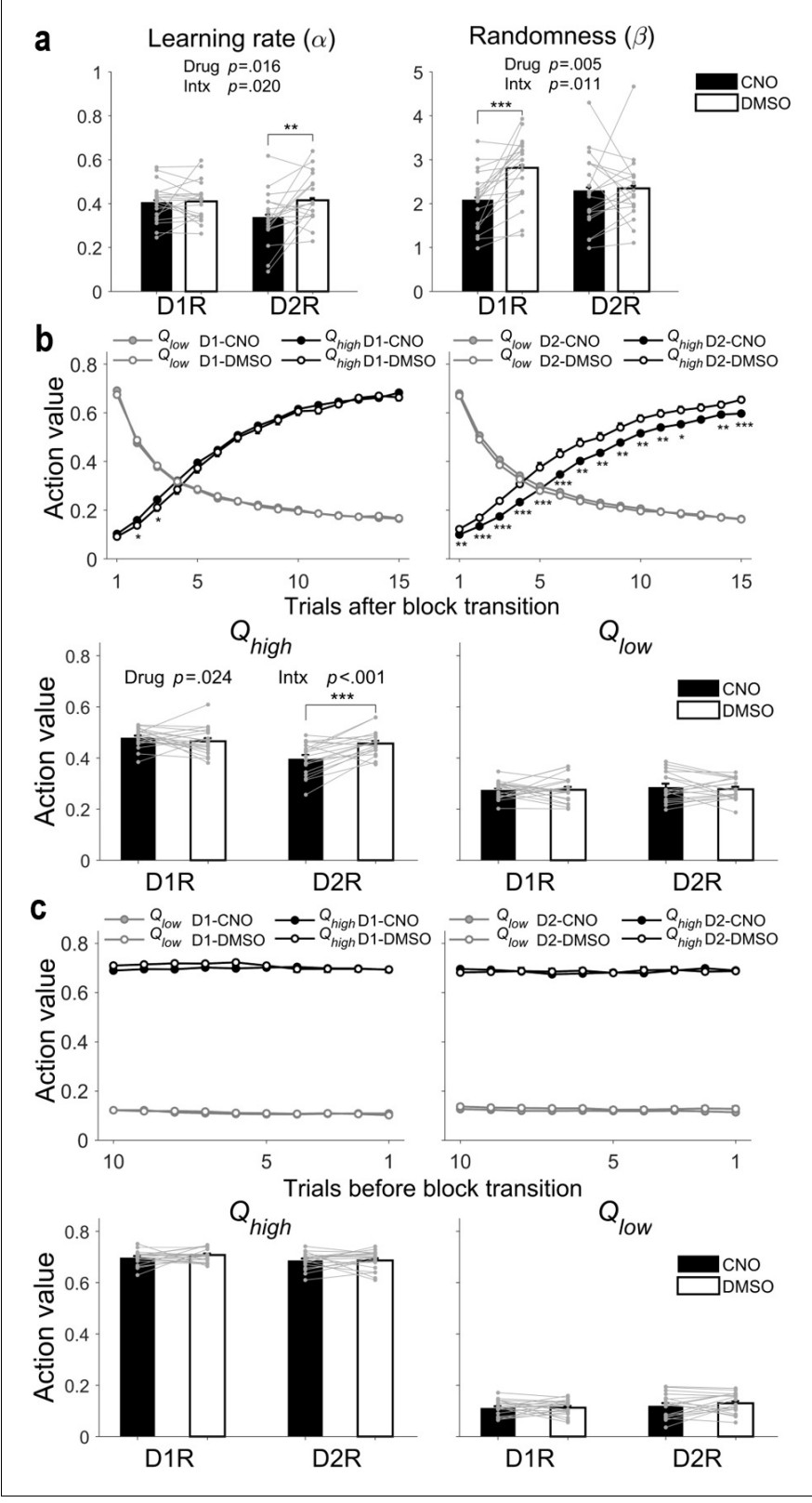

**Figure 4.** Effects of CNO on learning rate and randomness in action selection. (**a**) Learning rate (α) and randomness in action selection (β), estimated from behavioral data during the TAB task, were compared between DMSO and CNO sessions (means ± SEM across animals). (**b**) Top, trial-by-trial action values during the initial 15 trials after block transition. Bottom, mean (± SEM across animals) action values in the dynamic state. (**c**) Top, trial-by-trial action values during the last 10 trials of a block. Bottom, mean (± SEM across animals) action values in the steady state. Gray circles and connecting

*Figure 4 continued on next page*

*Figure 4 continued*

lines, individual animal data. Asterisks indicate the results of Bonferroni post-hoc tests (*p<0.05; **p<0.01; ***p<0.001) for those measures with significant mouse line × drug interaction (Intx) effects (two-way mixed-design ANOVA).

DOI: https://doi.org/10.7554/eLife.46050.008

The following figure supplement is available for figure 4:

**Figure supplement 1.** Consistent results were obtained using variants of the Q-learning model.
DOI: https://doi.org/10.7554/eLife.46050.009

choline acetyltransferase-positive striatal neurons are labeled in D1R-Cre and D2R-Cre mice (*Figure 1—figure supplement 1*; see also *Shin et al., 2018*). Previous theories on circuit operations of the basal ganglia have focused on relative contributions of direct and indirect pathways to controlling actions in the same domain (*Albin et al., 1989*; *Alexander and Crutcher, 1990*; *DeLong, 1990*; *Frank et al., 2004*; *Hikosaka et al., 2000*; *Kravitz and Kreitzer, 2012*; *Mink, 1996*; *Nambu, 2008*; *Soares-Cunha et al., 2016*). Our results raise the possibility that direct and indirect pathways play more important roles in different domains of decision making. One limitation of our study is the lack of precisely timed manipulation of striatal neuronal activity, which would be useful for gaining insight into the roles played by specific activity patterns of D1R- and D2R-expressing striatal neurons and their interactions with dopamine circuits in decision making. Future studies employing manipulation techniques that allow precisely timed manipulation of striatal neural activity, such as optogenetics, may provide useful information in this regard.

We have shown previously that value learning is impaired in D2R-knockout mice, but not D1R-knockout mice (*Kwak et al., 2014*). There also exists a large body of literature indicating a role for D2R in reversal learning, although D1R has also been implicated in this process (*Izquierdo et al., 2017*; *Klanker et al., 2013*; *Waltz, 2017*). Reversal learning is also impaired by selectively inactivating striatal neurons in the indirect pathway, but not the direct pathway (*Piray, 2011*; *Yawata et al., 2012*). Furthermore, indirect pathway striatal SPNs carry stronger previous reward signals than direct pathway SPNs in mice (*Shin et al., 2018*); and in monkeys, striatal injection of a D2R, but not D1R, antagonist impairs learning from past outcomes (*Lee et al., 2015*). These studies are consistent with the current findings, which suggest a critical role of the striatal indirect pathway in mediating value learning.

It has been proposed that direct- and indirect-pathway striatal neurons mediate learning from positive and negative outcomes, respectively (*Frank et al., 2007*; *Frank et al., 2004*). Our results are inconsistent with this proposal in that inactivating D1R-expressing striatal neurons had no significant effect on learning rate. Our results are also inconsistent with this proposal in that inactivating D2R-expressing striatal neurons impaired learning from positive outcomes. The effect of inactivating D2R-expressing striatal neurons on the action value for a high-reward–probability target ($Q_{high}$) was much greater than that for a low-reward–probability target ($Q_{low}$) after block transition. Furthermore, using models containing separate learning parameters for positive and negative outcomes ($\alpha_p$ and $\alpha_n$, respectively; models 3, 5 and 6), we found a significant reduction in $\alpha_p$, but not $\alpha_n$, following inactivation of D2R-expressing striatal neurons. These results suggest that D2R-expressing striatal neurons play a more important role in learning from positive than negative outcomes.

Dopamine has long been proposed to play a role in gain control and modulation of corticostriatal action selection processes (*Beeler et al., 2010*; *McClure et al., 2003*; *Servan-Schreiber et al., 1990*). In particular, a previous modeling study proposed that tonic dopamine regulates randomness in action selection via D1R-expressing, but not D2R-expressing, striatal neurons (*Humphries et al., 2012*), a suggestion consistent with our findings. A recent study has also shown that activating striatal direct and indirect pathways alters the gain of cortical motor commands (*Yttri and Dudman, 2016*). Our results also support roles of striatal neurons in gain control. Changing the randomness in action selection (β) is equivalent to changing the gain of value-dependent action selection without altering action values (see *Equation 2*). Likewise, changing learning rate (α) is equivalent to changing the gain of reward prediction error (RPE)-dependent learning (see *Equation 1*; note that $RPE = R(t) - Q_a(t)$). Our results suggest that striatal direct and indirect pathways may be involved in controlling the gain, not only of motor commands, but also of value-based decision making. Considering that stimulation of D1R- and D2R-expressing SPNs induces distinct patterns of responses in downstream

structures (*Lee et al., 2016*), inactivation of these SPN subtypes is likely to exert distinct effects on downstream structures as well. It remains to be determined how D1R- or D2R-expressing striatal neuronal inactivation affects downstream structures, such as the EP, SNr, thalamus, and motor cortical areas so as to compromise value learning or value-dependent action selection.

Our results are not entirely consistent with previous findings. We previously showed that both D1R- and D2R-expressing SPNs convey value and RPE signals, which would suggest their involvement in both value-dependent action selection and value-updating processes. In particular, the activity of D1R- and D2R-expressing SPN populations increases and decreases, respectively, as a function of value (*Shin et al., 2018*), which fits well with the antagonistic effects of striatal D1R versus D2R (or direct versus indirect pathway SPN) manipulations on reward-based learning (*Hikida et al., 2010*; *Kravitz et al., 2012*; *Nakamura and Hikosaka, 2006*; *Tai et al., 2012*; *Yawata et al., 2012*). Likewise, antagonistic effects of D1R- versus D2R-expressing SPN stimulation on motor behavior have been reported (*Durieux et al., 2012*; *Kravitz et al., 2010*; *Yttri and Dudman, 2016*). However, in the present study, inactivation of D1R- or D2R-expressing SPNs impaired two different aspects of value-based decision making. It may be that both direct and indirect pathways are involved in action selection and value learning, but D1R (or D2R)-expressing SPNs alone may be sufficient to support value-dependent action selection (or value updating), such that strong stimulation yields antagonistic effects whereas inactivation yields selective effects. Alternatively, the direct and indirect pathways may play selective roles, and seemingly antagonistic stimulation effects are because of indirect effects of strong, potentially non-physiological, stimulation. Note that direct and indirect pathway striatal neurons often exhibit activity that cannot be explained by a simple antagonistic or synergistic relationship between the two pathways (e.g., *Cazorla et al., 2014*; *Cui et al., 2013*; *Shin et al., 2018*). Likewise, direct and indirect pathway manipulations often lead to behavioral outcomes that cannot be readily explained by their antagonistic or synergistic actions (e.g., *Jin et al., 2014*; *Vicente et al., 2016*). Also note that we inactivated both the dorsomedial and dorsolateral striatum, which are likely to make substantially different contributions to behavioral control (*Balleine et al., 2009*; *Ito and Doya, 2011*; *Khamassi and Humphries, 2012*; *Yin and Knowlton, 2006*). Clearly, further studies are needed to make coherent sense of all these findings and to understand how striatal direct and indirect pathways work together to contribute to making optimal choices in a dynamic and uncertain environment.

## Materials and methods

**Key resources table**

| Reagent type (species) or resource | Designation | Source or reference | Identifiers | Additional information |
|---|---|---|---|---|
| Strain, strain background (*Mus musculus*) | STOCK Tg(Drd1-cre) EY217Gsat/Mmucd | Gene Expression Nervous System Atlas | RRID:MMRRC_030778-UCD | |
| Strain, strain background (*Mus musculus*) | STOCK Tg(Drd2-cre) ER44Gsat/Mmucd | Gene Expression Nervous System Atlas | RRID:MMRRC_017263-UCD | |
| Recombinant DNA reagent | AAV8-hSyn-DIO-hM4Di-mCherry | Addgene (PMID:21364278) | RRID:Addgene_44362 | |
| Recombinant DNA reagent | AAV2-hSyn-DIO-eGFP | Addgene | RRID:Addgene_50457 | |
| Chemical compound, drug | clozapine-N-oxide | TOCRIS | Cat. #:4936 | |
| Chemical compound, drug | dimethyl sulfoxide | TOCRIS | Cat. #:3176 | |
| Software, algorithm | Matlab 9.4 | Matworks | R2018a | |

### Subjects

C57BL/6J BAC transgenic mouse lines expressing Cre recombinase under control of dopamine D1R or D2R (*Drd1*-EY217 and *Drd2*-ER44, respectively) were obtained from Gene Expression Nervous

System Atlas. The animals were extensively handled and then water-deprived so that their body-weights were maintained at ~80% of ad libitum levels throughout the experiments. Each mouse was housed in an individual home cage, and all experiments were performed in the dark phase of a 12 hr light/dark cycle. A total of 31 D1R-Cre and 30 D2R-Cre mice were used for expression of h4DMi-mCherry in the striatum. Of these, 11 D1R-Cre and 11 D2R-Cre mice were tested in the reversal task only, 10 D1R-Cre and nine D2R-Cre mice were tested in the dynamic TAB task only, and 10 D1R-Cre and 10 D2R-Cre mice were tested in both the reversal and TAB tasks. The mice tested in the reversal task were assigned randomly to CNO- or DMSO-treatment groups. An additional five D1R-Cre and five D2R-Cre mice were used for expression of eGFP in the striatum and were tested in the reversal task only. Only male mice were used in the present study and all were 10–15 wk old at the time of virus injection surgery. All animal care and experimental procedures were performed in accordance with protocols approved by the directives of the Animal Care and Use Committee of Korea Advanced Institute of Science and Technology (approval number KA2018-08).

## Virus injection

Mice were anesthetized with isoflurane (1.0–1.2% [vol/vol] in 100% oxygen), and two burr holes were made bilaterally at 0.3 mm anterior and 2.0 mm lateral to bregma. AAV8-based, modified human M4 muscarinic receptor (AAV8-hSyn-DIO-hM4Di-mCherry; 31 D1R-Cre and 30 D2R-Cre mice) or AAV2-based enhanced green fluorescent protein (AAV2-hSyn-DIO-eGFP; five D1R-Cre and five D2R-Cre mice; Addgene) expression constructs were injected bilaterally at a depth of 3.0 mm from the brain surface at a rate of 0.05 μl/min (total volume, 2 μl). The injection needle was held in place for 15 min before and after the injection.

## Behavioral tasks

Animals were trained in self-paced instrumental learning tasks in an operant chamber (product #ENV-307A; MED Associates, Fairfax, VT, USA). The chamber was customized to contain three nose-poke holes, each with an infrared photobeam sensor for detecting a nose poke and an LED, on the front wall. A water-delivery nozzle was also located inside each of the left and right nose-poke holes (*Figure 2a*). Each animal was tested in a reversal task and/or a dynamic TAB task. In both tasks, the session began by turning on the central LED. A nose poke in the central hole turned off the central LED and turned on the LEDs on both sides. The animal was free to choose between the two lit nose-poke holes at this stage. A nose poke in either the left or right hole turned off the left and right LEDs, triggered water delivery (30 μl) in some trials (correct-choice trials in the reversal task and sto-chastically with a given probability in the TAB task) at the chosen target, and turned on the center LED. Mice were acclimated to the chamber on day 1 (free exploration of the chamber for 1 hr without reward delivery) and experienced shaping training on day 2 (center LED on - > nose poke - > center LED off and side LEDs on - > reward delivery on both sides; 60 trials or 1 hr) before being trained in the tasks.

The reversal task consisted of five stages with progressively increasing reversal frequency (*Kwak et al., 2014*) (one session per day). In the first stage, mice were trained to choose one target (either left or right; counterbalanced across animals) to obtain a water reward (30 μl). They per-formed 60 daily trials for 3 d. In the second stage, animals were trained to choose the opposite tar-get (the unrewarded target in stage 1) for 4 d (60 daily trials). In the third stage, the location of the rewarding target changed from that of the previous day (across-session reversal). Third-stage train-ing persisted for 4 d with 60 daily trials. In the fourth stage, in addition to changing the location of the rewarding target from that of the previous day, the location of the rewarding target was reversed midway through daily training (at trial 31; total daily trials, n = 60) for 4 d. In the final stage, in addition to changing the location of the rewarding target from that of the previous day, the loca-tion of the rewarding target was reversed twice during daily training (at trials 31 and 61; total daily trials, n = 90).

The dynamic TAB task consisted of four blocks of trials, each of which consisted of 35–50 trials (one session per day; 24 hr apart); DMSO and CNO were injected on alternate days, with the order of drug injection counterbalanced across animals. A total of 35, 40, 45 or 50 trials, determined ran-domly, were conducted per block (means ± SD: 38.8 ± 6.1 trials per block and 155.3 ± 16.0 trials per session). In each block, one target delivered water with a relatively high probability (72%) and the

other target delivered water with a relatively low probability (12%). The reward probabilities in the first block were determined randomly and were reversed across block transitions.

## Determination of dynamic and steady states

Dynamic and steady states were determined separately for each block as previously described (*Jeong et al., 2018*). Animal choice data were smoothed using a moving average of seven trials. The dynamic state lasted until the probability of choosing the higher-reward–probability target (P(H)) exceeded 70% of the maximum value after block transition. The steady state corresponded to the period from the trial at which P(H) exceeded 90% of the maximum value until the end of the block. The mean (± SD across animals) numbers of trials for the dynamic state were 7.5 ± 4.4 for D1R-DMSO, 7.6 ± 5.0 for D1R-CNO, 8.4 ± 5.0 for D2R-DMSO, and 8.5 ± 7.3 for D2R-CNO. For the steady state, means ± SD were 9.6 0 ± 6.1 for D1R-DMSO, 9.5 ± 6.8 for D1R-CNO, 10.5 ± 6.8 for D2R-DMSO, and 10.4 ± 7.0 for D2R-CNO.

## Reinforcement learning models

Animal choice behavior in the dynamic TAB task was analyzed using the Q-learning model (*Sutton and Barto, 1998*), in which action values in the $t^{th}$ trial ($Q_a(t)$) were updated, as follows:

$$\begin{aligned} \text{if } a = a(t), \quad & Q_a(t+1) = (1-\alpha)Q_a(t) + \alpha R(t) \\ \text{else} \quad & Q_a(t+1) = Q_a(t), \end{aligned} \tag{1}$$

where $a$ represents an action (left or right target choice), $R(t)$ denotes the reward (i.e., trial outcome) in the $t^{th}$ trial (1 if rewarded and 0 otherwise), and $\alpha$ indicates the learning rate. Action selection was determined using a softmax function of the difference in action values ($Q_L(t) - Q_R(t)$), as follows:

$$P_L(t) = \frac{1}{1 + \exp(-\beta(Q_L(t) - Q_R(t)))}, \tag{2}$$

where $P_L(t)$ is the probability of choosing the left goal and $\beta$ is the inverse temperature, which determines the degree of randomness in action selection (smaller $\beta$ values induce more random choices).

We also analyzed animal choice behavior using several variants of the Q-learning model (model 1, parameters, $\alpha$ and $\beta$) by adding additional parameters and using separate learning constants for positive and negative outcomes ($\alpha_{pos}$ and $\alpha_{neg}$, respectively). Model two had a choice bias ($V_L$) as an additional parameter (parameters, $\alpha$, $\beta$ and $V_L$). Model three had separate learning constants for positive and negative outcomes, and also included a choice bias (parameters, $\alpha_{pos}$, $\alpha_{neg}$, $\beta$ and $V_L$). Model four had a choice bias, win-stay (WS), and lose-switch (LS) as additional parameters (parameters, $\alpha$, $\beta$, $V_L$, WS and LS). Model five had separate learning constants for positive and negative outcomes and included a choice bias, win-stay, and lose-switch ($\alpha_{pos}$, $\alpha_{neg}$, $\beta$, $V_L$, WS and LS). All four models can be expressed by the following equations:

$$\begin{aligned} \text{if } a = a(t), \\ \text{if } R(t) = 1 \quad & Q_a(t+1) = (1-\alpha_{pos})Q_a(t) + \alpha_{pos}R(t) - \gamma\_win \\ \text{else} \quad & Q_a(t+1) = (1-\alpha_{neg})Q_a(t) + \alpha_{neg}R(t) - \gamma\_lose \\ \text{else} \\ & Q_a(t+1) = Q_a(t), \end{aligned} \tag{3}$$

where $\alpha_{pos}$ and $\alpha_{neg}$ are learning rates for rewarded and unrewarded trials, respectively, and $\gamma\_win$ and $\gamma\_lose$ are the penalty terms for repeating the same choice. Actions were chosen according to the softmax action selection rule, as follows:

$$P_L(t) = \frac{1}{1 + \exp(-\beta(Q_L(t) - Q_R(t)) + b)}, \tag{4}$$

where $b$ is a bias term for selecting the left target. The following constraints were applied to these parameters for models 2–4: model 2, $\alpha_{pos} = \alpha_{neg}, \gamma_{win} = \gamma_{lose} = 0$; model 3, $\gamma_{win} = \gamma_{lose} = 0$; model 4, $\alpha_{pos} = \alpha_{neg}$.

In addition, for model 6, we added terms for uncertainty-based exploration ($\varepsilon$ and $\rho$) (*Frank et al., 2009*; *Kwak et al., 2014*) to model 5 (parameters, $\alpha_{pos}$, $\alpha_{neg}$, $\beta$, $V_L$, WS, LS, $\varepsilon$ and $\rho$).

Details of the modeling are described in our previous paper (*Kwak et al., 2014*). Model parameters were estimated separately for each mouse and for each condition (DMSO or CNO injection) by pooling choice data of all sessions based on a maximum-likelihood procedure.

### Statistical analysis

Sample sizes were determined based on the sample sizes used in our previous study (*Kwak et al., 2014*), in which performances of D1R- and D2R-knockout mice were compared with those of wild-type mice in similar behavioral tasks as used in the present study. Two-way ANOVA and Bonferroni post-hoc tests were used for group comparisons. All statistical tests were two-tailed. A p-value<0.05 was used as the criterion for a statistically significant difference. Data are expressed as means ± SEM unless noted otherwise. The data were analyzed with Matlab software (The MathWorks, Inc, MA, USA). Raw data and code for reproducing this work are archived at Dryad (https://doi.org/10.5061/dryad.4c80mn5).

## Acknowledgements

We thank Namjung Huh for his help with data analysis and Jung Hwan Shin for his helpful comments on the initial manuscript. This work was supported by the Research Center Program of the Institute for Basic Science (IBS-R002-G1) (MWJ)

## Additional information

### Funding

| Funder | Grant reference number | Author |
|---|---|---|
| Institute for Basic Science | IBS-R002-G1 | Min Whan Jung |

The funders had no role in study design, data collection and interpretation, or the decision to submit the work for publication.

### Author contributions

Shinae Kwak, Conceptualization, Data curation, Formal analysis, Investigation, Methodology; Min Whan Jung, Conceptualization, Resources, Formal analysis, Supervision, Validation, Writing—original draft, Project administration, Writing—review and editing

### Author ORCIDs

Min Whan Jung (iD) https://orcid.org/0000-0002-4145-600X

### Ethics

Animal experimentation: The experimental protocol was approved by the Animal Care and Use Committee of the Korea Advanced Institute of Science and Technology (Daejeon, Korea; approval number approval number KA2018-08).

### Decision letter and Author response

Decision letter https://doi.org/10.7554/eLife.46050.015
Author response https://doi.org/10.7554/eLife.46050.016

## Additional files

### Supplementary files

• Supplementary file 1. Model comparison.
DOI: https://doi.org/10.7554/eLife.46050.010

• Transparent reporting form
DOI: https://doi.org/10.7554/eLife.46050.011

## Data availability

Data are available via Dryad under https://dx.doi.org/10.5061/dryad.4c80mn5.

The following dataset was generated:

| Author(s) | Year | Dataset title | Dataset URL | Database and Identifier |
|---|---|---|---|---|
| Kwak S, Jung MW | 2019 | Data from: Distinct roles of striatal direct and indirect pathways in value-based decision making | https://dx.doi.org/10.5061/dryad.4c80mn5 | Dryad Digital Repository, 10.5061/dryad.4c80mn5 |

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
