## [Decision Letter]

Thank you for submitting your article "Distinct roles of striatal direct and indirect pathways in value-based decision making" for consideration by *eLife*. Your article has been reviewed by three peer reviewers, including Geoffrey Schoenbaum as the Reviewing Editor and Reviewer #1, and the evaluation has been overseen by Ronald Calabrese as the Senior Editor. The following individual involved in review of your submission has agreed to reveal their identity: Mehdi Khamassi (Reviewer #2).

The reviewers have discussed the reviews with one another and the Reviewing Editor has drafted this decision to help you prepare a revised submission.

Summary:

In the current study, the authors test the role of the direct and indirect pathways in reversal learning in mice. D1 vs D2 neurons were inactivated via CNO during several variants of a reversal task. The results demonstrate clear reversal deficits overall, and modest differences in effects of inactivation of the two pathways. Modeling of the results suggested that the effects could be distinguished as reflecting learning versus the use of the new information.

Essential revisions:

Overall the reviewers agreed that the study addressed an important question and was well designed and executed. However the reviewers were ultimately split on whether the major novel conclusions – the differential effects of inactivation of D1 vs D2 networks – were robustly supported by the data. In particular the differential effects in Figure 3 were very modest and it was not clear that they would hold up to a more appropriate statistical analysis of the entire design (vs t-tests). This experiment also seemed to lack the CNO-only control for comparison. On the other hand, the modeling results were judged to be robust. As a result the novelty, and our invitation for a revision, rests largely on these modeling data.

So it is essential in any revision that concerns and questions in the review regarding these analyses be fully addressed, and that the writing of these analyses be made as accessible as possible. Ideally this might include a brief discussion of assumptions of these analyses. It is also important that the data in Figure 3 be analyzed appropriately. If the results are not statistically meaningful unless they are analyzed by a series of isolated t-tests, then this may be a problem on re-review. Finally some softening of the claims may help given the effect sizes; for instance is it certain that the functions are entirely dichotomous or does the dominance of one pathway *bias* the network toward one function versus the other?

*Reviewer #1:*

In this study, the authors test effects of inhibition of striatal D1 and D2 neurons on performance in a choice task in which the mouse picked between two nosepoke responses on each trial to obtain reward. In the first task, one nosepoke was rewarded while the other was not, and this reversed across blocks of trials, first across and then within sessions. In the second task, the nosepokes were associated with high (72%) and low (12%) probability of reward, and the probabilities reversed in 4 trial blocks each session. The authors found that inhibition of either neuron type (via DREADD/CNO) caused marked reversal deficits in the first task relative to CNO alone and more subtle deficits in the second task. An analysis of performance early versus late in the second task revealed that D2 inhibition affected behavior early in a block and D1 inhibition affected behavior later. The authors interpreted this via modeling as reflecting completely dissociable roles of the two neural subtypes in learning versus using information about action values.

Overall this study addresses an important question, which is the respective roles of these two neural subtypes and circuits in value-based behavior. However I think the results have a number of problems that need to be corrected. One is that the key second experiment lacks any control for non-specific effects of CNO. This lack is mitigated by the inclusion of this control in the first experiment, but the second experiment is quite different and contains the key effects. So the lack of this control is a problem I think. That the two groups show different effects in the second experiment may mitigate this issue, but possibly one effect is CNO alone and the other is specific? In any event, the lack of this comparison is a problem and will be a criticism of the study.

A second procedural problem for me is the statistical analyses. Most of the comparisons are pairwise t-tests I think, and the effects are quite small. Do these comparisons stand up when done with ANOVAs or similar, looking for interactions, etc?

Two other issues for me are more interpretive. One is that I do not really see the clear dichotomy between the phases of the task and the cognitive functions. Obviously it *could* be as the authors say, but I don't think it *must* be as the authors say. Indeed similar claims have been made in the past – Jones and Mishkin, 1972 for example argued that impairments after reversal prior to 50% performance reflected a failure to unlearn the old information whereas impairments after this point reflected a failure to learn the new information. Yet we know that old information can breakthrough days or weeks after it has apparently been unlearned (!). And new learning does not wait for unlearning. Likewise here it seems to me that a failure to adequately learn versus use information could appear as poor performance early or late, depending on other factors. So I think this is a potential interpretation of the finding but not the only one.

Another interpretive problem I have is that the authors used a manipulation that will affect activity constitutively, without regard to when the network is phasically active. Even though it is reversible, the effect of CNO will be across the entire trial and even session of course. The firing patterns of these neurons and particularly any interaction with dopamine circuits is obviously going to be time-varying. I think the conclusions should recognize this limitation.

Despite these concerns, I think the study is overall very nice and clearly provides novel findings bearing on a question of interest. I think if the above technical issues are addressable and the language softened a bit, it would make a nice report.

*Reviewer #2:*

The authors present the results of D1-Cre or D2-Cre injections in mice for selectively inactivating D1R- or D2R-expressing dorsal striatal neurons, respectively in two different tasks: a reversal task with progressively increasing reversal frequency and a dynamic two-armed bandit task. 6 different reward-based learning models were fitted to the data and compared in order to assess which computational mechanisms may best explain the experimental data. Results show that inactivation of D1R- and D2R-expressing striatal neurons selectively impairs value-dependent action selection and value learning, respectively.

The results are to my knowledge novel, timely, and shed new light on a debate about the selective roles of D1R- and D2R- expressing striatal neurons in reward-based learning. In particular, previous studies using optogenetic stimulations had found dissociated effects of D1R- and D2R- stimulation on positive versus negative reinforcement (e.g., Kravitz et al., 2012). In constrast, some other studies had shown a role of D2R- but not D1R- stimulation in reward-based learning (e.g., Lee et al., 2015). And a few theories have argued for a role of dopamine in the basal ganglia on gain control for action selection, distinct from its role in learning, which is in agreement with the present results. I thus think that the present results bring important new insights into the understanding of other dopamine-related functions in the striatum than the classical implication in reward-based learning.

Below, I raise a few questions and make a few suggestions to improve the manuscript.

Results final paragraph: '*Q_high_* was slightly higher during a few trials after block transition in CNO compared to DMSO sessions' -> Was this difference statistically significant?

In the Discussion, the authors try to argue against the possibility that 'D1R- and/or D2R- expressing striatal interneurons (GABAergic fast-spiking interneurons and cholinergic tonically-active neurons)' might have contributed to the observed results. They discard this possibility by saying that only small fractions of interneurons were labeled here. Nevertheless, the percentage of labeled neurons might not be the crucial parameter if a few interneurons can have a massive effect on performance. I am not sure what the authors mean when they write that the interneurons were labeled 'in similar degrees in the D1-Cre and D2-Cre mice'. I guess they mean non-different percentages of labeled interneurons between the two types of mice. If this is the case, first the authors should apply some statistical test to verify that there are no differences in proportions (e.g., chi-square proportion test). Second, this does not answer the question whether a small percentage of labeled interneurons may have had a strong contribution (at least not a negligible contribution compared to that of MSN neurons). Is there a way the authors can answer this question?

In the Discussion, I think the authors should also emphasize that a central theory (Frank et al., 2004) argues not only that D1R- and D2R- expressing striatal neurons both contribute to learning, but also that D1R-ones are involved in positive learning (GO) and D2R-ones are involved in negative learning (NO-GO). Here, the results show that D2R-expressing neurons are involved in both positive and negative learning. e.g., model fitting with distinct learning rates for positive and negative feedback show significant variations in both for D2-CNO compared to D2-DMSO (Figure 4—figure supplement 1). The same figure even shows that for the 2 models with higher degree complexity, it is only the positive learning rate which is affected for D2. I guess this is more complicated to interpret given the number of free parameters and possible interactions between parameters. Nevertheless, all this could be more extensively discussed in the manuscript.

The authors discuss the possible effects on action selection in terms of gain control by the basal ganglia over cortical motor commands (Yttri and Dudman, 2016, 2018). I think that to be fair, the authors should cite the first paper (to my knowledge) which proposed a role for dopamine in gain control: Servan-Schreiber, Printz and Cohen, 1990. Later, a computational model of incentive salience integrated a role of dopamine signals in the striatum in two different functions: reinforcement learning and gain control in a softmax function to decide whether to GO or not: McClure, Daw and Montague, 2003. However, this model did not extend the mechanism to decision-making between multiple actions. Later on, a theory of the role of dopamine in the basal ganglia on the exploration-exploitation trade-off was proposed, in direct extensions of this theoretical framework: Humphries, Khamassi and Gurney, 2012. In this theory, tonic dopamine directly modulates the inverse temperature (parameter β) in the softmax function for action selection. Importantly, Importantly, through neural network simulations, the latter theory predicted that tonic dopamine should regulate the exploration-exploitation trade-off specifically via D1R-expressing striatal neurons, not D2R-ones. The present results are in perfect accordance with this theory and bring further insights on the functional dissociation between D1R- and D2R-expressing striatal neurons.

In subsection “Reinforcement learning models”, in order to better appraise the different mechanisms in the compared computational models, it is important to show the equations for each of them. For instance, I bet that the choice bias VL, and win-stay lose-shift parameters WS and LS play a role via the softmax equation 2. But it is better to make this explicit by showing how the equations change for each of the tested models.

*Reviewer #3:*

In this paper, the authors aimed to determine how direct and indirect pathway MSNs work in tandem to influence choice behavior. The manuscript was well written, and the experiments were executed thoroughly. They show that inhibiting D1 or D2 MSNs disrupted reversal learning (Figure 2), which was the largest effect they observed. The data looks very convincing on this point.

Upon parsing their data further (Figure 3 and 4), the authors report a difference in DREADD mediated inhibition of D2 vs D1-MSNs in early vs late trials per block, respectively. The authors interpret this as a difference in "value based action selection" (early trials) vs. "value based updating" (late trials). I was not convinced that this operational definition was reasonable, as it seemed a bit arbitrary to define the early trials as one component of the decision making process and the late trials as another. To make such claims the authors would likely need to utilize different tasks that specifically engaged each process as independently as possible.

Beyond this issue, the difference in each pathway that they observed in this part of the paper were very small, on the order of 5-10% changes in behavior, and trending in the same direction (impaired performance with CNO) in both phases, for both groups (Figure 3). From the figure legend, it appears that the authors analyzed this data with multiple paired t-tests instead of ANOVAs, which is not appropriate in a multiple comparison design like this. My suspicion is that if the data were analyzed by ANOVA it would reveal no effect of group between these measures.

Overall, I thought there was far more similarity than differences in the effects of manipulating each pathway, and found a dissonance between the data and the language used in the paper, which sought to highlight differences, concluding that "D1R- and D2R-expressing striatal neurons selectively impairs value-dependent action selection and value learning, respectively." Though the data is promising and raises interesting questions, it alone is insufficient to conclude that D2 MSN activity underlies value-based learning and D1 underlies action-selection.

As it currently stands, this work presents an exciting initial phase of a project but I believe it is insufficient for publication as a stand-alone manuscript in *eLife*.

[Editors’ note: this article was subsequently rejected after discussions between the reviewers, but the authors were invited to resubmit after an appeal against the decision.]

Thank you for submitting your work entitled "Distinct roles of striatal direct and indirect pathways in value-based decision making" for consideration by *eLife*. Your article has been reviewed by three peer reviewers, including Geoffrey Schoenbaum as the Reviewing Editor and Reviewer #1, and the evaluation has been overseen by a Senior Editor. The following individual involved in review of your submission has agreed to reveal their identity: Mehdi Khamassi (Reviewer #2).

Our decision has been reached after consultation between the reviewers. Based on these discussions and the individual reviews below, we regret to inform you that your work will not be considered further for publication in *eLife*.

This decision was reached by consensus across the three reviewers, in reading the revision and during the interactive discussion. The main problem was that the key statistical comparisons did not support the dissociation that was the basis of the modeling and the heart of the paper. Two of the three reviewers noted this as an essential revision on evaluating the original manuscript, and on discussion of the revision, all three reviewers agreed that the issue had not been corrected in the revision. While the modeling was judged to be interesting, it was not argued to be enough of an advance on its own to publish without clear biological support for the claimed dichotomy in the function of the D1 and D2 systems. And while additional subjects might be added to correct this shortcoming, the policy at *eLife* is not to request additional experiments of prolonged duration. We are very sorry for the negative outcome.

*Reviewer #1:*

I was disappointed in the revisions I am afraid. The key issue is that the statistical assessment on Figure 3 did not support the claimed dissociation that lead to the modeling. The bidirectional effects are exceedingly small in the two directions and without full statistical support for the dissociation, I think it becomes questionable whether the biological system even shows the effect that is modeled.. Additionally the other concerns were addressed in relatively marginal ways. This did not help, but was not a determining factor. If the statistics were strong and the dissociation was robust, I would have found it acceptable I think. But I think the paper really hinges on this comparison. I am very sorry that I cannot be more positive.

*Reviewer #2:*

The authors have addressed all my concerns.

*Reviewer #3:*

I thank the authors for being responsive to the reviewer comments, and for re-analyzing the data in Figure 3. In my prior review I had two main concerns:

1) The authors analyzed the data in Figure 3 with many paired t-tests instead of an ANOVA.

2) I found the conclusions about differences between the two pathways to be over-stated, as they were based on relatively small effects and an improper statistical analysis.

The authors were responsive to the statistical concern, and now analyze data in Figure 3 with ANOVAs. This confirmed that several of their prior conclusions on these pathways were not significant when analyzed this way. In this reanalysis, they first analyzed the entire behavioral dataset with four individual ANOVAs (with no correction for multiple comparisons). They now find no significant interaction between drug x mouse line for the proportion of rewarded trials (P(R)), higher-reward-probability target choices (P(H)), probability of win-stay (P(WS)). They did detect a significant interaction between drug x mouse line for the probability of lost-stay (P (LS), p=0.019). Without a prior hypothesis pointing them to lose-stay behavior, I think it would be reasonable to correct their α, which limits my enthusiasm for the interaction with p=0.019. If they performed a Bonferoni correction to avoid Type I error, their α would be 0.0125. Even with a more lenient correction it is likely that this interaction would be on the border of significance.

They continue to parse the behavioral data into early and late trials. Here, they now perform 8 separate ANOVAs (same as above but done separate for early and late trials), again with no correction for multiple comparisons. In the early (dynamic) phase, they found a significant drug x mouse line interaction in P(R) (p=0.04), but not on any other measure. In the late (static) phase, there were significant drug x mouse line interactions on P(H) (p=0.029) and P(LS) (p=0.005). Based on the number of analyses performed on the same underlying data, some α correction should be used. If they performed a Bonferoni correction to avoid Type I error, their α would be 0.0042 and none of these would be considered significant. If they used a correction method that was less aggressive against Type I error, it is unlikely that any beyond the P(LS) in the dynamic phase would be significant.

Based on these new analyses, my first concern stands, in that the prior reporting of differences was due in part to the use of inappropriate statistical analysis. Unfortunately this is still the case, as no corrections were made for multiple comparisons when running several ANOVAs on different aspects of the same dataset. Based on this new statistical analysis and interpretation, I still find the conclusions of the paper to be over-interpreted. The Abstract and title highlight the differences between these pathways, when in almost every condition inhibition of either pathway caused a similar pattern of behavioral impairment.

One way I would be convinced that there's a force greater than chance at work here would be if the authors performed a permutation analysis on this same dataset. If they randomize which group each animal belongs to and re-run their 12 ANOVAs, how many ANOVAs show a significant interaction between drug x mouse line?

---

## [Author Response]

Essential revisions:Overall the reviewers agreed that the study addressed an important question and was well designed and executed. However the reviewers were ultimately split on whether the major novel conclusions – the differential effects of inactivation of D1 vs D2 networks – were robustly supported by the data. In particular the differential effects in Figure 3 were very modest and it was not clear that they would hold up to a more appropriate statistical analysis of the entire design (vs t-tests). This experiment also seemed to lack the CNO-only control for comparison. On the other hand, the modeling results were judged to be robust. As a result the novelty, and our invitation for a revision, rests largely on these modeling data.

*So it is essential in any revision that concerns and questions in the review regarding these analyses be fully addressed, and that the writing of these analyses be made as accessible as possible. Ideally this might include a brief discussion of assumptions of these analyses. It is also important that the data in Figure 3 be analyzed appropriately. If the results are not statistically meaningful unless they are analyzed by a series of isolated t-tests, then this may be a problem on re-review. Finally some softening of the claims may help given the effect sizes; for instance is it certain that the functions are entirely dichotomous or does the dominance of one pathway* bias *the network toward one function versus the other?*

We addressed these legitimate concerns in our revised manuscript. Please see our responses to individual comments below. Briefly, we replaced t-tests with two-way ANOVA, discussed the CNO-control issue, and softened our claim on specific roles of direct-and indirect-pathway neurons.

Reviewer #1:

[…] Overall this study addresses an important question, which is the respective roles of these two neural subtypes and circuits in value-based behavior. However I think the results have a number of problems that need to be corrected. One is that the key second experiment lacks any control for non-specific effects of CNO. This lack is mitigated by the inclusion of this control in the first experiment, but the second experiment is quite different and contains the key effects. So the lack of this control is a problem I think. That the two groups show different effects in the second experiment may mitigate this issue, but possibly one effect is CNO alone and the other is specific? In any event, the lack of this comparison is a problem and will be a criticism of the study.

As the reviewer noted, that CNO selectively affects learning rate in D1-Cre mice and randomness in action selection in D2-Cre mice argues against non-specific effects of CNO. A non-specific effect of CNO, if any, should be observed in both animal groups (e.g., learning rate is lowered in both animal groups). We also obtained consistent results in our previous study. CNO impaired learning rate in only one of four different animal groups expressing hM4Di in different hippocampal subregions, while having no significant effect on learning rate or randomness in action selection in the remaining three animal groups during a dynamic two-armed bandit task (Jeong et al., 2018).We briefly discussed this matter in the revised manuscript.

A second procedural problem for me is the statistical analyses. Most of the comparisons are pairwise t-tests I think, and the effects are quite small. Do these comparisons stand up when done with ANOVAs or similar, looking for interactions, etc?

We replaced t-tests with two-way ANOVAand obtained largely similar results. Two-way ANOVA yielded similar conclusions for the analyses shown in Figure 2 and 4. Regarding Figure 3, as expected, we failed to obtain significant interaction effects for some measures that showed selectivity in CNO effect between D1-Cre and D2-Cre mice upon t-tests. Nevertheless, we found CNO effects that are selective between D1-Cre and D2-Cre mice for some behavioral measures. We found that CNO significantly decreases P(R) in D2-Cre, but not D1-Cre, mice in the dynamic state, and significantly decreases P(H), P(WS) and P(LS) in D1-Cre, but not D2-Cre, mice in the steady state (revised Figure 3).These results support the possibility that D1R-and D2R-expressing striatal neurons contribute differently to the neural processes underlying the animal’s choice behavior.

*Two other issues for me are more interpretive. One is that I do not really see the clear dichotomy between the phases of the task and the cognitive functions. Obviously it* could *be as the authors say, but I don't think it* must *be as the authors say. Indeed similar claims have been made in the past – Jones and Mishkin, 1972 for example argued that impairments after reversal prior to 50% performance reflected a failure to unlearn the old information whereas impairments after this point reflected a failure to learn the new information. Yet we know that old information can breakthrough days or weeks after it has apparently been unlearned (!). And new learning does not wait for unlearning. Likewise here it seems to me that a failure to adequately learn versus use information could appear as poor performance early or late, depending on other factors. So I think this is a potential interpretation of the finding but not the only one.*

We fully agree with the reviewer’s comment, and we by no means meant to equate early-and late-phase inactivation effects with deficits in value learning and value-based action selection, respectively. Value learning and value-dependent action selection are presumably always at work. What we try to argue is that effects of manipulating value-learning vs. value-dependent action selection processes on the animal’s choice behavior would be relatively more pronounced during early and late trials after block transition, respectively. To avoid misunderstanding, we revised the text related to Figure 3 as the following:

“To further explore this possibility, we separately analyzed animal choice behavior in the dynamic and steady states (early and late trials after block transition, respectively; see Materials and methods) between which relative contributions of value-updating and value-dependent action-selection processes to choice behavior are likely to vary.”

We also revised the text related to Figure 4 as the following:

“Differences in the pattern of CNO effects on animal choice behavior during dynamic and steady states between D1-Cre and D2-Cre mice raises the possibility that D1R-and D2R-expressing striatal neurons may contribute differently to the neural processes underlying value-based decision making.”

Another interpretive problem I have is that the authors used a manipulation that will affect activity constitutively, without regard to when the network is phasically active. Even though it is reversible, the effect of CNO will be across the entire trial and even session of course. The firing patterns of these neurons and particularly any interaction with dopamine circuits is obviously going to be time-varying. I think the conclusions should recognize this limitation.

We agree and discussed this matter in the revised text.

Despite these concerns, I think the study is overall very nice and clearly provides novel findings bearing on a question of interest. I think if the above technical issues are addressable and the language softened a bit, it would make a nice report.

Thank you for these positive comments. We de-emphasized our claim on specific roles of direct and indirect pathway neurons throughout the revised manuscript.

Reviewer #2:

[…] The results are to my knowledge novel, timely, and shed new light on a debate about the selective roles of D1R- and D2R- expressing striatal neurons in reward-based learning. In particular, previous studies using optogenetic stimulations had found dissociated effects of D1R- and D2R- stimulation on positive versus negative reinforcement (e.g., Kravitz et al., 2012). In constrast, some other studies had shown a role of D2R- but not D1R- stimulation in reward-based learning (e.g., Lee et al., 2015). And a few theories have argued for a role of dopamine in the basal ganglia on gain control for action selection, distinct from its role in learning, which is in agreement with the present results. I thus think that the present results bring important new insights into the understanding of other dopamine-related functions in the striatum than the classical implication in reward-based learning.

Thank you for these positive comments.

Below, I raise a few questions and make a few suggestions to improve the manuscript.Results final paragraph: 'Q_high_ was slightly higher during a few trials after block transition in CNO compared to DMSO sessions' -> Was this difference statistically significant?

*Q_high_* was significantly higher in CNO sessions compared with DMSO sessions in the second and third trials after block transition in D1-Cre mice(Bonferroni post-hoc tests following two-way repeated measures ANOVA). This is indicated with asterisks in Figure 4B (left, line graph).

In the Discussion, the authors try to argue against the possibility that 'D1R- and/or D2R- expressing striatal interneurons (GABAergic fast-spiking interneurons and cholinergic tonically-active neurons)' might have contributed to the observed results. They discard this possibility by saying that only small fractions of interneurons were labeled here. Nevertheless, the percentage of labeled neurons might not be the crucial parameter if a few interneurons can have a massive effect on performance. I am not sure what the authors mean when they write that the interneurons were labeled 'in similar degrees in the D1-Cre and D2-Cre mice'. I guess they mean non-different percentages of labeled interneurons between the two types of mice. If this is the case, first the authors should apply some statistical test to verify that there are no differences in proportions (e.g., chi-square proportion test). Second, this does not answer the question whether a small percentage of labeled interneurons may have had a strong contribution (at least not a negligible contribution compared to that of MSN neurons). Is there a way the authors can answer this question?

We agree that a small percentage of labeled interneurons may exert relatively strong influences on behavior. As suggested, we performed Fisher’s exact test and found no significant difference in the number of labeled interneurons between D1-Cre and C2-Cre mice (see Figure 1—figure supplement 1). In fact, their proportions were very similar (PV, 2.7and 2.3%, respectively; ChaT, 6.3 and 6.4%, respectively). Hence, any effect of inactivating labeled interneurons on the animal’s behavior would have been similar between D1-Cre and D2-Cre mice. Given the reviewer’s comment, we revised the related text from “This is unlikely because only small fractions of parvalbumin-positive (< 3%) and choline acetyltransferase-positive (< 7%) striatal neurons are labelled and in similar degrees in the D1-Cre and D2-Cre mice” to “However, this is unlikely because similar, small percentages of parvalbumin-positive and choline acetyltransferase-positive striatal neurons are labelled in D1-Cre and D2-Cre mice”.

In the Discussion, I think the authors should also emphasize that a central theory (Frank et al., 2004) argues not only that D1R- and D2R- expressing striatal neurons both contribute to learning, but also that D1R-ones are involved in positive learning (GO) and D2R-ones are involved in negative learning (NO-GO). Here, the results show that D2R-expressing neurons are involved in both positive and negative learning. e.g., model fitting with distinct learning rates for positive and negative feedback show significant variations in both for D2-CNO compared to D2-DMSO (Figure 4—figure supplement 1). The same figure even shows that for the 2 models with higher degree complexity, it is only the positive learning rate which is affected for D2. I guess this is more complicated to interpret given the number of free parameters and possible interactions between parameters. Nevertheless, all this could be more extensively discussed in the manuscript.

As suggested, we discussed this matter in the revised text. Please note that new analysis results (two-way ANOVA) indicate significant effect of D2R neuronal inactivation on only positive learning rate in all models tested.

The authors discuss the possible effects on action selection in terms of gain control by the basal ganglia over cortical motor commands (Yttri and Dudman, 2016, 2018). I think that to be fair, the authors should cite the first paper (to my knowledge) which proposed a role for dopamine in gain control: Servan-Schreiber, Printz and Cohen, 1990. Later, a computational model of incentive salience integrated a role of dopamine signals in the striatum in two different functions: reinforcement learning and gain control in a softmax function to decide whether to GO or not: McClure., Daw and Montague, 2003. However, this model did not extend the mechanism to decision-making between multiple actions. Later on, a theory of the role of dopamine in the basal ganglia on the exploration-exploitation trade-off was proposed, in direct extensions of this theoretical framework: Humphries, Khamassi and Gurney, 2012. In this theory, tonic dopamine directly modulates the inverse temperature (parameter β) in the softmax function for action selection. Importantly, Importantly, through neural network simulations, the latter theory predicted that tonic dopamine should regulate the exploration-exploitation trade-off specifically via D1R-expressing striatal neurons, not D2R-ones. The present results are in perfect accordance with this theory and bring further insights on the functional dissociation between D1R- and D2R-expressing striatal neurons.

Yes, it would be fair to cite these literatures. They are now cited and discussed in the revised text.

In subsection “Reinforcement learning models”, in order to better appraise the different mechanisms in the compared computational models, it is important to show the equations for each of them. For instance, I bet that the choice bias VL, and win-stay lose-shift parameters WS and LS play a role via the softmax equation 2. But it is better to make this explicit by showing how the equations change for each of the tested models.

Done as suggested except the last model (model 6).The model 6 contains terms for uncertainty-based exploration (ɛ and p)which require somewhat lengthy descriptions. Given that CNO effects on uncertainty-based exploration are not the main focus of our study and that the model 6 is fully described in our previous paper (Kwak et al., 2014), we referred to previous papers (Frank et al., 2009; Kwak et al., 2014) instead of elaborating all the related equations.

Reviewer #3:

In this paper, the authors aimed to determine how direct and indirect pathway MSNs work in tandem to influence choice behavior. The manuscript was well written, and the experiments were executed thoroughly. They show that inhibiting D1 or D2 MSNs disrupted reversal learning (Figure 2), which was the largest effect they observed. The data looks very convincing on this point.Upon parsing their data further (Figure 3 and 4), the authors report a difference in DREADD mediated inhibition of D2 vs D1-MSNs in early vs late trials per block, respectively. The authors interpret this as a difference in "value based action selection" (early trials) vs. "value based updating" (late trials). I was not convinced that this operational definition was reasonable, as it seemed a bit arbitrary to define the early trials as one component of the decision making process and the late trials as another. To make such claims the authors would likely need to utilize different tasks that specifically engaged each process as independently as possible.

We agree with the reviewer’s comment. Reviewer #1 also raised a similar concern. Please see our response to the third comment of reviewer #1.

Beyond this issue, the difference in each pathway that they observed in this part of the paper were very small, on the order of 5-10% changes in behavior, and trending in the same direction (impaired performance with CNO) in both phases, for both groups (Figure 3). From the figure legend, it appears that the authors analyzed this data with multiple paired t-tests instead of ANOVAs, which is not appropriate in a multiple comparison design like this. My suspicion is that if the data were analyzed by ANOVA it would reveal no effect of group between these measures.

We replaced t-tests with two-way ANOVAas suggested. As the reviewer predicted, we failed to obtain significant interaction effects for some measures that showed selectivity in CNO effect between D1-Cre and D2-Cre mice upon t-tests. Nevertheless, we found CNO effects that are selective between D1-Cre and D2-Cre mice in some behavioral measures. CNO significantly decreases P(R) in D2-Cre, but not D1-Cre, mice in the dynamic state, and significantly decreases P(H), P(WS) and P(LS) in D1-Cre, but not D2-Cre, mice in the steady state. As we indicated in our response to the third comment of reviewer #1, value updating and value-dependent action selection are presumably always at work. Hence, effects of manipulating value-learning vs. value-dependent action-selection processes on the animal’s choice behavior would be relatively more pronounced during early and late trials after block transition, respectively, rather than being exclusive. Hence, deficits in these processes are expected to influence choice behavior during early and late trials in the same direction. Also, the probabilistic nature of the task (rather than correct vs. incorrect choices) is likely to have contributed to overall small effect sizes when the animal’s behavior (rather than its underlying processes) is examined (Figure 3). In addition, partial inactivation of striatum, which is a large structure, may have contributed to overall small effect sizes. Nevertheless, CNO effects were quite consistent across animals so that we could find significant and selective CNO effects on some measures upon pairwise comparisons (repeated-measures ANOVA).

Overall, I thought there was far more similarity than differences in the effects of manipulating each pathway, and found a dissonance between the data and the language used in the paper, which sought to highlight differences, concluding that "D1R- and D2R-expressing striatal neurons selectively impairs value-dependent action selection and value learning, respectively." Though the data is promising and raises interesting questions, it alone is insufficient to conclude that D2 MSN activity underlies value-based learning and D1 underlies action-selection.

Please note that our model-based analysis yielded a clear double dissociation in the effects of CNO treatment on learning rate and randomness in action selection(Figure 4). It is true that there are similarities between D1 and D2 MSN inactivation effects on behavioral measures(Figure 3). Note, however, that changes in two different underlying processes can lead to similar changes in the final end product (i.e., choice behavior).Conversely, differential effects of two manipulations on the final end product indicate differential effects of manipulation on the underlying processes. Considering that the task is probabilistic in nature (small manipulation effect) and value learning and value-based action selection are presumably always operating(relatively rather than absolutely stronger/weaker manipulation effects at early vs. late trials), it is remarkable that differential effects of D1 vs. D2 MSN inactivation could be detected even by examining the final end product (behavioral variables).

[Editors’ note: the author responses to the re-review follow.]

The main criticism during the second round of review is that our statistical test results in Figure 3 do not support our claimed dissociation of inhibition effects between D1-Cre and D2-Cre mice. Specifically, the reviewer #3 pointed out that the lack of Bonferroni correction for multiple ANOVAs is problematic. The reviewer #3 then suggested a permutation test as an alternative way of testing the significance of our results.

We agree this is a legitimate concern. Even though we showed significant interaction effects in multiple ANOVAs (3 out of 8 ANOVAs; 4 out of 12 ANOVAs if we include all-trial analysis results), we did not test whether this is significant. We consulted a biostatistician and were advised to use the permutation test the reviewer #3 suggested. We were told that correcting for multiple comparisons would be inappropriate for our data set (insufficient statistical power to reach a stringent threshold). Instead, testing the likelihood of finding significant interaction effect in 3 or more ANOVAs would a legitimate way of testing the significance of our results. As suggested by the reviewer #3, we randomly assigned D1-cre and D2-mice into two animal groups and repeated the same analysis (total 8 ANOVAs; P(R), P(H), P(WS) and P(LS) in the dynamic and steady states). Out of 100 such permutations, we found no case in which significant animal group*drug interaction effect was found in 3 or more ANOVAs (c.f., we found 3 cases in which significant interaction effect was found in 2 ANOVAs). Similar results were obtained when we performed a permutation test for 12 ANOVAs including all-trial data; we found no case in which significant animal group*drug interaction effect was found in 4 or more ANOVAs out of 100 permutations (c.f., we found 1 case in which significant interaction effect was found in 3 ANOVAs). These new results indicate that our results are unlikely to be obtained by chance.

Given that the new analysis results address the main criticism, we are wondering whether you would be willing to re-consider your decision. We believe that it would be quite beneficial to share our data with the neuroscience community and stimulate further studies along this line.

Reviewer #1:

I was disappointed in the revisions I am afraid. The key issue is that the statistical assessment on Figure 3 did not support the claimed dissociation that lead to the modeling. The bidirectional effects are exceedingly small in the two directions and without full statistical support for the dissociation, I think it becomes questionable whether the biological system even shows the effect that is modeled.. Additionally the other concerns were addressed in relatively marginal ways. This did not help, but was not a determining factor. If the statistics were strong and the dissociation was robust, I would have found it acceptable I think. But I think the paper really hinges on this comparison. I am very sorry that I cannot be more positive.

As we explained in the cover letter and the response to the comment of reviewer #3, we performed the permutation test suggested by the reviewer #3 and obtained results indicating that our findings in Figure 3 are highly significant. We agree that the effects shown in Figure 3 are small. However, this is not very surprising because 1) the task is probabilistic in nature, 2) value learning and value-based action selection are presumably always operating (relatively rather than absolutely stronger/weaker manipulation effects at early vs. late trials), and 3) the striatum is a large structure and we only partially inactivated striatum. Given that CNO effects on overall P(R) and P(H) (all-trial analysis) were very similar between D1-Cre and D2-Cre mice, it is remarkable that differential effects of D1 vs. D2 neuronal inactivation could be detected on behavioral measures when we divided trials into early (dynamic state) and late (steady state).

Reviewer #3:

[…] They continue to parse the behavioral data into early and late trials. Here, they now perform 8 separate ANOVAs (same as above but done separate for early and late trials), again with no correction for multiple comparisons. In the early (dynamic) phase, they found a significant drug x mouse line interaction in P(R) (p=0.04), but not on any other measure. In the late (static) phase, there were significant drug x mouse line interactions on P(H) (p=0.029) and P(LS) (p=0.005). Based on the number of analyses performed on the same underlying data, some α correction should be used. If they performed a Bonferoni correction to avoid Type I error, their α would be 0.0042 and none of these would be considered significant. If they used a correction method that was less aggressive against Type I error, it is unlikely that any beyond the P(LS) in the dynamic phase would be significant.Based on these new analyses, my first concern stands, in that the prior reporting of differences was due in part to the use of inappropriate statistical analysis. Unfortunately this is still the case, as no corrections were made for multiple comparisons when running several ANOVAs on different aspects of the same dataset. Based on this new statistical analysis and interpretation, I still find the conclusions of the paper to be over-interpreted. The Abstract and title highlight the differences between these pathways, when in almost every condition inhibition of either pathway caused a similar pattern of behavioral impairment.One way I would be convinced that there's a force greater than chance at work here would be if the authors performed a permutation analysis on this same dataset. If they randomize which group each animal belongs to and re-run their 12 ANOVAs, how many ANOVAs show a significant interaction between drug x mouse line?

Many thanks for elaborating issues related to statistics and suggesting an alternative way of testing the significance of our results. We consulted a biostatistician and were told that the permutation test you suggested would be a legitimate way of testing the significance of our results. As suggested, we randomly assigned D1-cre and D2-mice into two animal groups and repeated the same analysis (total 8 ANOVAs; P(R), P(H), P(WS) and P(LS) in the dynamic and steady states). Out of 100 such permutations, we found no case in which significant animal group x drug interaction effect was found in 3 or more ANOVAs (c.f., we found 3 cases in which significant interaction effect was found in 2 ANOVAs). This new result is described in the revised text (L 218). Results were similar when we performed a permutation test for 12 ANOVAs (additional 4 ANOVAs for P(R), P(H), P(WS) and P(LS) for all trials); we found no case in which significant animal group x drug interaction effect was found in 4 or more ANOVAs out of 100 permutations (c.f., we found 1 case in which significant interaction effect was found in 3 ANOVAs). This indicates that our results are unlikely to be obtained by chance. Regarding your comment on the interaction effect in P(LS) (all-trial analysis), we agree that it is at best a weak effect and therefore deleted the related text in the revised manuscript (see new Figure 3).